# Paternal transmission of migration knowledge in a long-distance bird migrant

Patrik Byholm [1,5✉], Martin Beal [2,6], Natalie Isaksson [2,3], Ulrik Lötberg [4] & Susanne Åkesson [2]

While advances in biologging have revealed many spectacular animal migrations, it remains poorly understood how young animals learn to migrate. Even in social species, it is unclear how migratory skills are transmitted from one generation to another and what implications this may have. Here we show that in Caspian terns *Hydroprogne caspia* family groups, genetic and foster male parents carry the main responsibility for migrating with young. During migration, young birds stayed close to an adult at all times, with the bond dissipating on the wintering grounds. Solo-migrating adults migrated faster than did adults accompanying young. Four young that lost contact with their parent at an early stage of migration all died. During their first solo migration, subadult terns remained faithful to routes they took with their parents as young. Our results provide evidence for cultural inheritance of migration knowledge in a long-distance bird migrant and show that sex-biased (allo)parental care en route shapes migration through social learning.

[1] Novia University of Applied Sciences, Raseborgsvägen 9, FI-10600 Ekenäs, Finland. [2] Department of Biology, Lund University, Ecology Building, SE-223 62 Lund, Sweden. [3] Environmental Research Institute, North Highland College, University of the Highlands and Islands, Ormlie Road, Thurso KW14 7EE, UK. [4] BirdLife Sweden, Stenhusa Gård SE-386 62, Mörbylånga, Sweden. [5] Present address: Organismal & Evolutionary Biology, University of Helsinki, PO Box 65, FI-00014 Helsinki, Finland. [6] Present address: Avenida Minas Gerais 2, 2780-206 Oeiras, Portugal. ✉email: patrik.byholm@helsinki.fi

The seasonal migrations of many species of mammals, fish, reptiles, and birds span vast geographical scales across the globe[1–4]. While migrating, animals often move in groups ranging in size from just a few individuals to tens of thousands[5]. Joining a group of conspecifics may provide multiple benefits, including improved navigation capacity, foraging opportunities, and predator detection[6,7]. In particular, the migrations performed by gregarious birds are a well-known phenomenon in nature which have long fascinated human observers[2]. While in some bird species, individuals may end up in flocks by coincidence, being funnelled together by weather and geography[8], others travel in highly-structured social groups, often consisting of family units within larger flocks[2,9]. In such species, social learning is an important way in which naïve young acquire migratory skills[10,11], and conservation projects have successfully used learning behaviour to teach migration routes to inexperienced migrants by leading them with ultralight aircraft[11]. In social species, learning from experienced individuals is thus fundamental to the development of migratory performance, and often overrides innate genetically inherited preferences for migratory tendency and orientation[10,11].

Due to the practical challenge of following specific individuals relying on field-based observation alone, there is limited understanding of how animals migrating together actually interact. Therefore, it remains largely unknown the extent to which migratory skills are transmitted obliquely (from any experienced individual in the population to any naïve individual) and/or vertically (from parents to offspring) from one generation to the next[12]. In contrast to our understanding of migration in some mammals[4,13], it is also not known if male and female bird parents have distinct roles in this context or what is the cultural inheritance of migration. That is, if social learning and/or teaching[14,15] affects whether young of long-distance bird migrants repeat the migrations they experienced when migrating with informed adults as young in consecutive years.

To study how migratory knowledge is transferred between generations in a species which typically migrates solo or in family groups[16], we used GPS-devices to track 31 Caspian Terns *Hydroprogne caspia* between the breeding grounds in northern Europe and the wintering grounds in Africa during 2016–2020. We tracked adults and young from the same family groups to explore the importance of within-family interactions during the migration of these birds, and simultaneously gain insight into how extended parental care affects the survival of young birds during migration. Finally, to test whether this system fits the definition of cultural inheritance of migratory knowledge, i.e., that early-life experiences influence decisions later in life, we quantified the degree to which subadults making their first solo migrations used the same routes and stopover sites they visited when accompanied by a parent.

## Results

**Interactions between family members on migration.** Terns initiated autumn migration between July 15 and August 24 (Table 1, Supplementary Table 1). All young that survived to initiate migration migrated together with an adult bird upon leaving the breeding grounds. A total of 9 young terns migrated with their male parent (in 7 cases the male parent migrated with a single young and in one case a male parent migrated with two young), one young migrated with a male foster parent and one young migrated with a female parent. The observed male:female-ratio among parents migrating with young (10:1) was strongly male biased ($\chi^2 = 8.33$, $df = 1$, $p = 0.004$). One young tern that was abandoned on the breeding islet (Young 2, Family group 3) and another (Young 2, Family group 5) that lost contact with its

parent within 24 h of leaving the breeding area were both predated by white-tailed eagles *Haliaeetus albicilla* within days. Similarly, soon after initiating migration, two young (Young 1, Family group 2; Young 1, Family group 4) were predated by a northern goshawk *Accipiter gentilis* and a white-tailed eagle, respectively, while temporarily left unattended by their parents en route (Supplementary Table 1).

Breeding partners never migrated together (Fig. 1, Supplementary Fig. 1) and initiated migration on average $14.7 \pm 9.9$ (mean ± s.d.) days apart, with no significant difference between the sexes in terms of which sex initiated migration first (pairwise *t*-test, $t = 0.89$, $df = 8$, $p = 0.40$). Consequently, parent-young pairs and solo migrants from the same family groups did not differ in the onset of migration (Table 1). During migration, all surviving young migrated together with the parent until reaching the wintering grounds (Fig. 1, Supplementary Fig. 1, Supplementary Movie 1). Consequently, travel speed did not differ between adults and young of the same parent-young pair (pairwise t-test, $t = -0.39$, $df = 4$, $p = 0.72$), and at the population level, adults and young did not differ in their travel speeds (Table 1). However, solo-migrating parents migrated with a significantly higher travelling speed than did parent-young pairs (Table 1). Because young terns spent more time roosting (and less time foraging) than adults at stopover sites (GLMM, $b = 0.05 \pm 0.02$, $t = 2.70$, $p = 0.01$), adults and young migrating together spent significantly less time foraging while present at stopover sites than solo adults (Table 1).

**Breakup of the bond between parents and young.** Adults and young migrating together remained close to one another (pairwise distance: $\Delta D$ $0.6 \pm 2.0$ km (mean ± s.d.). Supplementary Movie 1) until the migration event was disrupted due to mortality, tag failure or the pair arrived successfully on the wintering grounds. Among the successful pairs, information on the breakup of the parental bond was available for 4 pairs from three different family groups. After arriving on the wintering grounds, the adult and young gradually started to spend more time apart ($\Delta D$: $1.1 \pm 3.7$ km) and after spending $68.8 \pm 17.5$ days together at the destination the bond eventually broke between late-October and early-December in all pairs ($\Delta D$: $333.7 \pm 661.0$ km) as inferred from segmented regressions of inter-individual distances (Fig. 2). After separation, the young either remained at the same wintering location as the adult ($n = 2$) or continued migrating further south ($n = 2$; one of which was the case of the young which followed a foster parent) (Supplementary Fig. 1).

**Fidelity to migration routes and stopover sites.** Adult and subadult Caspian terns showed strong fidelity to autumn migration routes across years (Fig. 3) more often selecting the same stopover sites in multiple seasons instead of unique ones each year (Chi-square test for equal proportions, $\chi^2 = 4.50$, $df = 1$, $p = 0.03$). Of the 32 registered stops, 69% ($n = 22$) took place at stopover sites visited in both seasons. However, the 62% ($n = 8/13$) rate of stopover site re-use observed among adults alone did not deviate from 1:1 ($\chi^2 = 0.69$, $df = 1$, $p = 0.41$), and consequently, the overall result is largely due to subadults re-using stopovers they visited on their first autumn migration ($\chi^2 = 4.26$, $df = 1$, $p = 0.04$). Of 19 stops recorded among young/subadults, 74% ($n = 14$) occurred at stopover sites used during both the first and second autumn migrations (Fig. 3). The lower re-use of stopover sites among adults migrating alone as compared to parent-young pairs is likely influenced by the larger distances between stopover sites observed among lone adults compared to among pairs of parents and young (Table 1).

**Table 1 Migration metrics and the results of GLMMs (Family ID specified as random intercept) as compared between pairs of parents and young and solo-migrating birds from the same family group.**

| Metric | Parent-young pair | Alone | | | | |
|---|---|---|---|---|---|---|
| | Avg ± s.d. | Avg ± s.d. | Estimate ± s.e. | *df* | *p* | Model# |
| Departure (1 = July 1st) | 32 ± 14 | 29 ± 13 | −5.47 ± 4.92 | 16.72 | 0.282 | #1 |
| Roosting (days) | 62 ± 30 | 51 ± 23 | −16.26 ± 11.78 | 18.89 | 0.184 | #2 |
| Distance between stopovers | 983 ± 170 | 1437 ± 653 | **454.40 ± 213.55** | **15.84** | **0.049** | #3 |
| Prop. time foraging on stopovers | 0.20 ± 0.06 | 0.25 ± 0.04 | **0.04 ± 0.02** | **19** | **0.024** | #4 |
| Arrival (1 = July 1st) | 94 ± 32 | 95 ± 26 | −1.30 ± 11.85 | 16.02 | 0.914 | #5 |
| Travel distance (km) | 4578 ± 1453 | 5328 ± 1375 | 779.07 ± 708.96 | 13.48 | 0.291 | #6 |
| Active travelling time (days) | 4 ± 1 | 4 ± 1 | 0.06 ± 0.47 | 18.80 | 0.894 | #7 |
| Travel speed (km/h) | 46.4 ± 5.5 | 52.8 ± 4.8 | **9.10 ± 2.19** | **16.75** | **<0.001** | #8 |

The results of model #4 is based on residual values (Methods), but average values are presented as calculated from actual data for easier direct interpretation. Statistically significant differences as judged from two-sided testing are in bold text. *Df*:s vary due to different sample size among models (Methods).

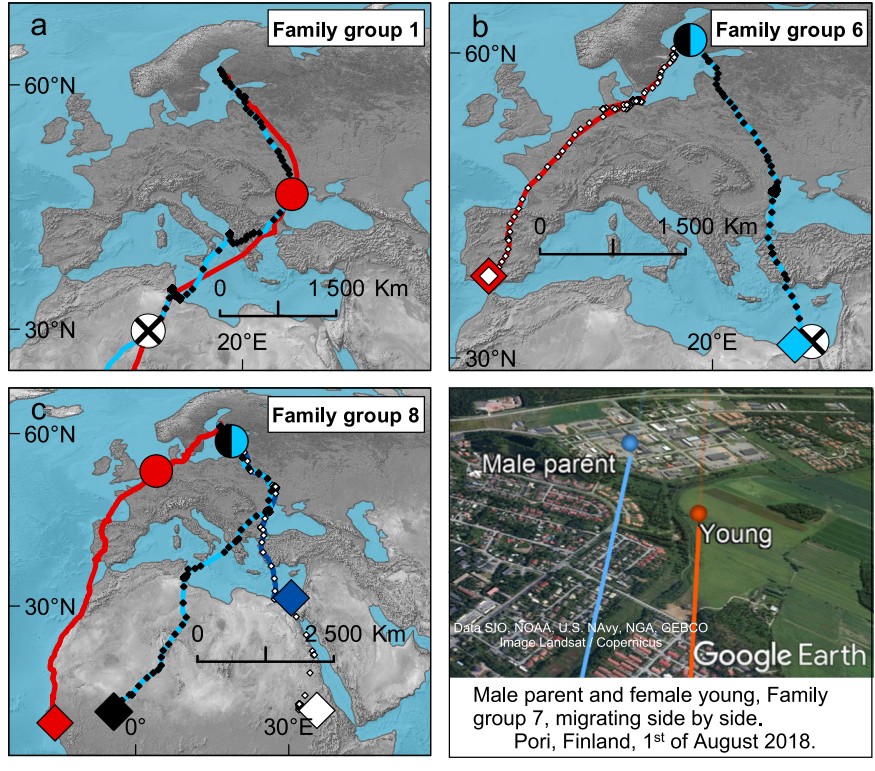

**Fig. 1 Examples of the migration routes of three Caspian tern family groups in the autumns of 2017–2019.** Migration tracks of adults are illustrated as continuous lines (red: Female parent, light blue: Male parent, dark blue: Foster male parent), that of young as diamonds (black: Young 1, white: Young 2). Large intact/split circles indicate geographical locations of tracked birds (as identified by their respective colours) when **a**, single, or **b** and **c**, the latter young (Young 2) from families of two young initiated migration together with the parent, i.e. they indicate the location of the other family member(s) when the last members from the same family group initiated migration. Crossed circles indicate locations of death/tag failure. Large diamonds indicate geographical location of birds on December 31st. For more details on symbology and family groups, see Supplementary Table 1 and Supplementary Fig. 1. A snapshot of an animation of real data (Supplementary Movie 1) showing the male parent and young from Family group 7 migrating side by side over Pori, Finland, is included in the bottom right-hand panel.

## Discussion

Previous work on how sociality shapes the migration of naïve individuals has been conducted on species migrating in large social groups[4,11]. However, in such systems it is challenging to distinguish the role(s) group members have in transmitting migratory knowledge to naïve individuals. We show that in Caspian terns, a species typically migrating solo or in small groups[16], breeding partners do not migrate together and that male (and foster male) parents carry most of the responsibility for guiding naïve young during their first outbound migration. The bond between a parent and young only gradually breaks down upon arrival to the wintering area.

The finding that adult males are largely responsible for accompanying naïve young migrants on their first outbound migration in a bird species providing bi-parental care is a finding that deserves further attention. It is unknown how widespread such paternal effects in migratory behaviour is among birds[17], but female offspring desertion and male-biased parental care is widespread in birds[18], including terns[19], and possibly originates in a sexual conflict where female parents gain in terms of residual

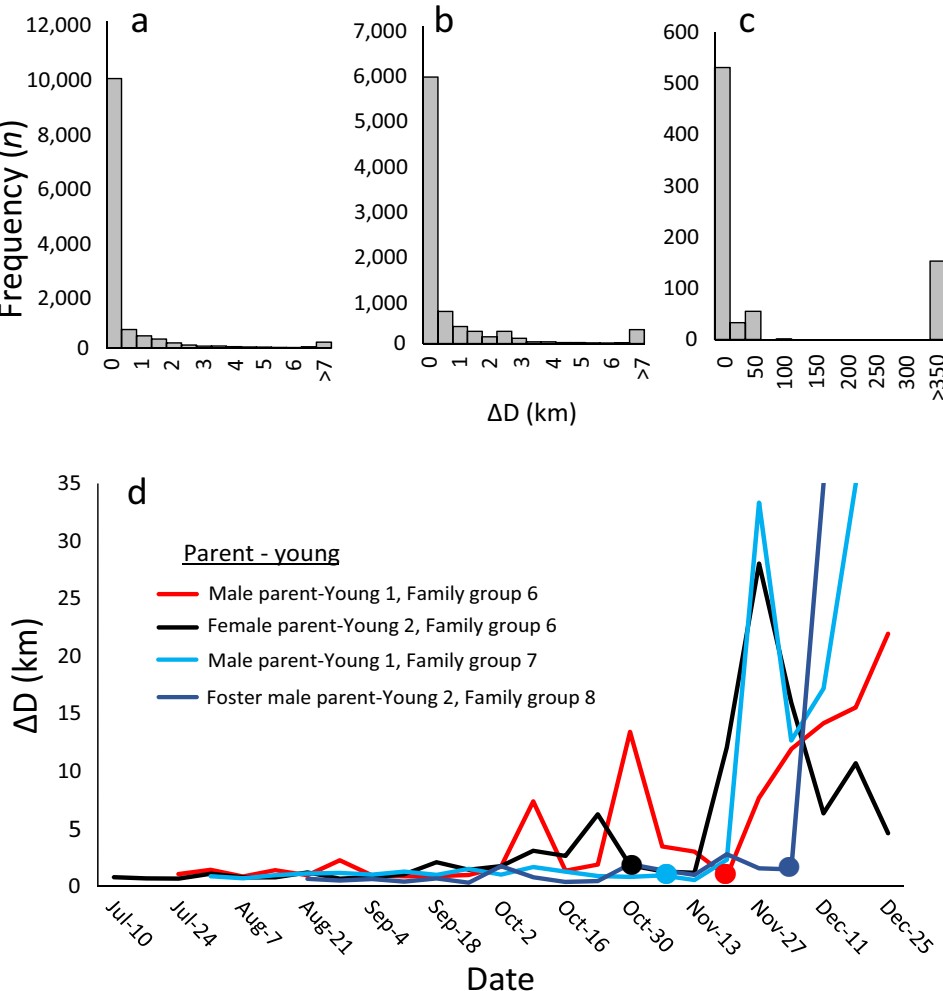

**Fig. 2 Bonds between experienced adult Caspian terns and naïve young during migration and after arrival on the wintering grounds.** Histograms show the distribution of pairwise distances (ΔD) between the adult and the young migrating together from **a**, the onset of migration until arrival in the wintering area ($n = 12,249$), **b** in the wintering area until the breakup of the parental bond ($n = 8242$), and **c** after the breakup of the parental bond ($n = 773$). **d** The average weekly distance between the parent and young (family and ID of parent-young is given in the inserted legend, cf. Supplementary Table 1) from onset of migration until December 31st among pairs successfully arriving in the wintering area ($n = 4$). Filled-in circles indicate the breakup date (mean ± s.e.) for each pair as judged from segmented regressions (Male parent-Young 1, Family group 6: 25-Nov ±3 days ($p < 0.0001$); Female parent-Young 2, Family group 6: 30-Oct ±5 days ($p < 0.0001$); Male parent-Young 1, Family group 7: 12-Nov ±2 days ($p < 0.0001$); Foster male parent-Young 2, Family group 8: 6-Dec ±0 days ($p < 0.0001$)).

fitness over time[20]. However, the fact that all the young terns which lost contact with their parent died from predation (Supplementary Table 1) does not lend direct support to this idea. Population studies which record lifetime reproductive success and individual survivorship are needed to reveal the ultimate explanation as to why female parents only occasionally migrate together with their young.

Given that Caspian terns, like some other birds[21–23], re-visit stopover sites along the same migration routes they followed in previous years (Fig. 3), the extension of parental care throughout the migratory period will likely have long-term impacts on the fitness of juvenile birds. In Caspian terns, the benefits of male parental care extend thousands of kilometres from the breeding grounds and suggests social learning is a key factor shaping the migratory behaviour of young. The observed differences in migration metrics (Table 1) between parent-young pairs and solo-migrating adults from the same families also indicate that parental care during migration impacts the migratory behaviour of adult birds. Similar to earlier work showing that birds in flocks fly slower than solo birds[24], we found that solo-migrating adults

travel faster than parent-young pairs, and travel greater distances between stopovers compared to parent-young pairs. Accompanying young appears to slow down parents during periods of active flight. Differences in migration speed and foraging behaviour during stopovers as observed between parent-young pairs and solo adults suggest migrating with young may come with a cost to the parent. According to the most accepted criteria defining teaching behaviour, i.e. that (1) the teacher modifies their behaviour in the presence of a naive observer, (2) there is a cost incurred by doing so, and (3) the teacher's modified behaviour leads the observer to acquire the behaviour faster or more efficiently than it might have done otherwise[25,26], our results suggest that adult Caspian terns accompanying young on migration constitute an example of teaching behaviour[14,15]. However, since there are no differences in terms of overall active travelling time, travel distance, nor the timing of arrival at the winter quarters between solo adults and parent-young pairs (Table 1), cost disparities appear to reset at the end of the migratory journey. This suggests that any possible costs associated with accompanying young on migration may be of relatively short-term nature,

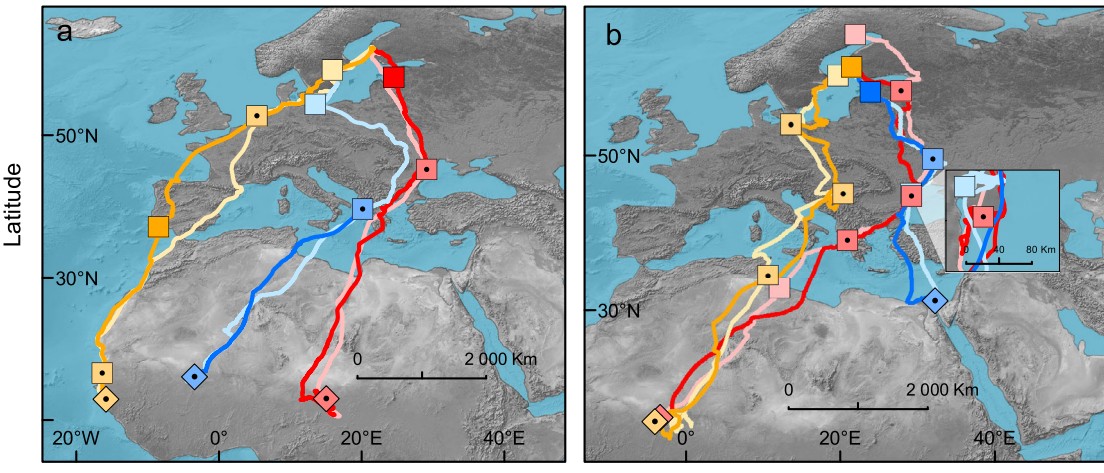

**Fig. 3 Repeated use of autumn migration routes and stopover sites of young/subadult and adult Caspian terns in two consecutive years during 2017–2020.** Migration routes (n = 12) are depicted with lines, mean locations of stopover sites (n = 32) with squares, mean locations of wintering sites (n = 6) with diamonds. For both routes and stopover sites, a dark colour shade refers to the most recent year, whereas a light shade refers to a more distant year. Squares/diamonds with a black central dot and of an intermediate shade indicate stopover sites/wintering sites that were used in multiple years (twice), simple squares sites that were used in only one year. **a** Repeat migration routes and stopover site use of one young/subadult (blue) and two adults (red, yellow). The information from the young is from 2018 (year of birth) and 2020 (first year of [partial] autumn migration – bird returned only halfway to Eastern Europe), that of the adults depicted in red and yellow (female parents from family groups 4 and 8, respectively; Supplementary Table 1) are from 2018 and 2019. **b** Repeat migration routes and stopover site use of two young/subadult (red, yellow) and one adult (blue). The information from the young depicted in red are from 2018 (year of birth) and 2020 (first year of autumn migration), that of the young depicted in yellow are from 2017 (year of birth) and 2020 (first year of autumn migration) and that of the adult in blue (foster male parent from family group 8, Supplementary Table 1) are from 2018 and 2019.

calling for longitudinal investigations of the lifetime fitness effects of deciding whether or not to accompany young.

Our finding that young Caspian terns can join un-related adult males on migration when leaving the breeding site demonstrates that migration routes in addition to being vertically transmitted from the biological parents to offspring are occasionally transmitted as a result of fostering behaviour[27]. To the extent that transgenerational social transmission of migration behaviour is adaptive, such behavioural plasticity can be expected to impact the array of migration routes within genetic lineages. In contrast to findings in some mammals[28], social learning and cultural inheritance does, however, not seem to have led to fixed group-level cultures in Caspian terns given how much variation there is in routes (Supplementary Fig. 1). In several cases, when naïve young after becoming independent broke away from the parent at advanced stages of migration, the young continued to migrate to other (more distant) wintering sites (Figs. 1c and 2, Supplementary Fig. 1) using highly goal-oriented flight behaviour. Although genetically inherited programs[29] may contribute, the observed movements made by young without their parents strongly suggests they joined with other birds on these flights[11]. In a species like the Caspian tern that aggregates at staging areas during migration[16], there would typically be experienced individuals present that naïve birds could join. That the bond between the parent and young gradually breaks apart over more than two months upon arrival in the wintering quarters indicates that gaining survival skills and new contacts takes a long time after arriving on the wintering grounds. Taken together, the extended parental care and combination of vertical and oblique (from foster parents) pathways for migration knowledge transfer will have consequences for the migratory routes and stopovers used at the population level. Given that Caspian terns are consistent in their use of socially-transmitted migration routes, this system constitutes an example of cultural inheritance of migratory knowledge[12,30–33].

By allowing for rapid adjustments in migratory habits, the dynamic transgenerational social transfer of migratory knowledge we observed in Caspian terns may alleviate them of the initial selection pressures on suboptimal timing, routes, and behaviours observed in other migratory bird species[34]. Such behavioural plasticity may be particularly beneficial in highly variable environments or when resources are scattered in space and time[35,36]. Similarly, such a developmental bias[33] can be expected to be of relevance for the process of species range expansion. The fact that Caspian terns have a disjunct, yet widespread global species range[16] may partially be explained by the manner in which migration knowledge is rapidly transmitted across generations. Thus, whether migratory species can persist in the face of global climate change and widespread habitat loss[37] can be expected to depend in part on how effectively knowledge of successful migratory routes and stopover sites is transmitted from experienced to naïve individuals. Considering the recent widespread declines of migratory birds[38], there is an urgent need to improve our general understanding of how social learning contributes to flexibility in migratory strategies, to ultimately be able to take appropriate management actions to conserve migratory species across the world.

## Methods

**Ethics statement**. Permits to trap, take blood samples and deploy GPS-tracking devices on Caspian terns in Finland were issued by the Regional State Administrative Agency for Southern Finland (ESAVI/1068/04.10.07/2017) and the Centre for Economic Development, Transport and the Environment in Southwestern Finland (VARELY/875/2017). In Sweden permits were issued by Malmö-Lunds djurförsöksetiska nämnd (M470-12, M72-15, M74-20) and by the Swedish Environmental Protection Agency and the Swedish Ringing Office (NV-03567-16). The Centre for Economic Development, Transport and the Environment in Southern Ostrobothnia (EPOELY/1830/215) and Norrland County Administration Board (521-4026-18) issued permits to work in protected areas. Permit to store biological samples (blood/DNA) for scientific purposes in Finland was issued by the Centre for Economic Development, Transport and the Environment in Southwestern Finland (VARELY/875/2017).

**Field protocol**. Caspian terns were caught at breeding sites along the Finnish west coast (62°14′N, 21°17′E) and the Swedish east coast (57°16′N, 16°37′E and 65°18′N, 22°23′E). Adults ($n = 13$) were caught at the nest during the last week of the incubation phase in late May using spring nets, and juveniles ($n = 18$) were caught by hand from the breeding islet just prior to fledging. Solar-powered GPS-trackers (Ecotone GPS-GSM or OrniTrack GPS-GSM/GPRS trackers) were attached with a leg-loop harness using Teflon ribbon[39]. Tags weighed 18–20 g corresponding to c. 3% of tern's body mass at the time of deployment (595 ± 74 g; mean ± s.d.). The amount and type of data the devices delivered varied (from 5 min to several hours between GPS-fixes) depending on device model and programming schedule as well as voltage level. Sex was determined from DNA as extracted from blood samples using the salt extraction method[8].

**Demarcation of tracking data**. A total of 33 migration events (made up of 434,632 GPS-fixes) conducted by 27 (of the originally 31) terns that successfully initiated migration were studied in analyses on migration[40]. The initiation of migration was defined from when the tracked bird left the breeding area in continuous directional flight southwards at a typical travel speed (35.3 ± 8.3 km/h) for a distance of 116.5 ± 79.3 km in one stretch without returning to the breeding area that season (Supplementary Table 1). For the purpose of this study, the time window considered was defined from when the first member of a family group departed the breeding islet for migration until the end of the calendar year (or until the individual died/the tracker failed).

The tracking data was segmented into periods of roosting, foraging and directional flight behaviour using the relative position of the original GPS-fixes (hereafter 'fixes') and the speed calculated as the quotient of the linear distance and time between consecutive pairs of fixes. If the flight speed between ≥3 consecutive fixes was <10 km⁻ʰ, fixes were assigned to the intermediary class "resting". Resting fixes located <30 km from each other and where the bird spent >24 h were aggregated into a minimum convex polygon; resulting polygons were considered stopover sites. Fixes located inside a polygon or <5 km from the polygons' outer boundary with a calculated flight speed of ≥2 km⁻ʰ were classified as foraging behaviour at the stopover area, and otherwise classified as roosting behaviour. Of the remaining fixes (flight speed ≥10 km⁻ʰ), those located >20 km from identified stopover sites were classified as migration flight behaviour.

To be able to study when the parental bond broke up, tracking data of pairs of parental birds and young successfully reaching the wintering area were merged into one data set and sorted chronologically. Acknowledging that the internal clocks of the GPS-trackers are not likely to be in perfect synchrony, pairwise (parent-young) distances were calculated between fixes ≤20 min apart in time (ΔD) and used as a measure of the tightness of the bond between the adult and young. Even though this measure therefore is likely to overestimate real pairwise distances, this measurement error remains unchanged in time enabling studies of how the tightness of the parental bond changes with advancing season. In analyses comparing differences in migration behaviour between parent-young pairs and adults migrating alone (cf. Table 1), only data covering the time-window between departure from the breeding area and arrival in the wintering areas was considered. The deterioration of the bond between parents and young was determined using segmented regressions (excluding ΔD > 100 km for Family group 7 [Dec-18 – Dec-31] and Family group 8 [Dec-6 – Dec-31] to allow model convergence) where ΔD was set as dependent and date as explanatory variables. Because the average GPS sampling rate differed among tracking devices and measures of the proportion of time foraging were positively biased with GPS sampling rate (linear regression, $F = 13.59$, $df = 19$, $p = 0.002$), the residuals from the model between GPS sampling rate and the proportion of time spent foraging at roosts were used in model #4 (Table 1) instead of the raw data. In Table 1 the degrees of freedom vary between models, e.g., because some birds failed to arrive to the wintering area or because the status of adult birds (migrating together with young or migrating alone) changed en route when chicks died.

Repeat use of migration routes and stopover sites during the autumn migration event was inferred from 12 GPS-tracks (of which 4 are also part of the family tracking material [Supplementary Table 1],[40]) delivered by trackers carried by adult and young/subadult Caspian terns in two consecutive seasons. For adults, tracking data spans the time window from when birds left the breeding islet until having spent ≥10 days in the wintering area. Tracking data for first autumn migration of naïve migrants was delimited in the same way, but because they did not breed as subadults in their first year of return, the geographical location of the last night roost used before the initiation of autumn migration was used as a starting point of the migration event. When analysing re-use of stopover sites, minimum convex polygons used by the same bird in consecutive years <15 km apart were considered to represent the same stopover site.

**Statistical analyses**. Initial handling and sorting of GPS-tracking data as well as construction of maps was performed with ArcMap 10.3.1 and MS Excel 2019. All statistical analyses are two-sided and were conducted in R 4.0.3[41]. Where statistical models involved multiple individuals from the same family group (fit by REML and linear link in lme4 package), family group ID was entered as a random intercept to account for the non-independence of response variables in GLMMs. The break-up of the bond between parents and young was inferred from segmented regressions

using the segmented-package in R. The animated movie in Supplementary Movie 1 was made in Google Earth Pro 7.3.3.7786.

**Reporting summary**. Further information on research design is available in the Nature Research Reporting Summary linked to this article.

## Data availability
Detailed metadata on tracked birds part of family units and birds studied in analyses of repeat use of migration routes and stopover sites is available in Supplementary Table 1 and the Movebank Data Repository under Movebank ID 1480583194 (https://doi.org/10.5441/001.1.352qf1cv)[40]. The GPS-tracking data used in this study are available in the Movebank Data Repository under Movebank ID 1480583194 (https://doi.org/10.5441/001.1.352qf1cv)[40].

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

## Acknowledgements

Jaana Kekkonen sexed the terns from DNA-samples. Funding to P.B. was provided by The Society of Swedish Literature in Finland (grants: 1677, 2170, 2882), Swedish Cultural Foundation in Finland (grants: 108220, 117778, 137166, 240073). S.Å. was supported by a project grant from the Swedish Research Council (2016-03625), and from the Centre for Animal Movement Research (CAnMove) funded by a Linnaeus grant from the Swedish Research Council (349-2007-8690) and Lund University. Funding to U.L. was provided by Alvin's fund, Lindberg's Foundation, Petra Lundberg's Foundation, WWF Sweden, BirdLife Sweden and by County administration boards in Östergötland and Norrbotten counties.

## Author contributions

P.B. and S.Å. conceived the study, M.B., N.I., P.B. and U.L. conducted field work, P.B. analysed the data and wrote the manuscript with input from M.B., N.I. and S.Å.

## Competing interests

The authors declare no competing interests.
