## [Peer Review File · Nature Communications]

Paternal transmission of migration knowledge in a long-distance bird migrantEditorial Note: This manuscript has been previously reviewed at another journal that is not operating a transparent peer review scheme. This document only contains reviewer comments and rebuttal letters for versions considered at Nature Communications.

Reviewers' Comments:

Reviewer #2:

Remarks to the Author:

In my (Referee #2) comments on the submission pre-dating the present one I had said that "This article offers what I judge are two interlinked 'firsts' that argue for a positive recommendation for publication in this journal" and I outlined those two, but then added a moderately substantive list of my reservations, largely concerning interpretations and citations to appropriate literature. This can all be seen in the Response to Referees document accompanying the present submission.

The rationale for my earlier positive recommendation stands. I judge that the authors have done a thorough and helpful job of responding to all of my earlier requests for revisions and I am satisfied with the resultant, present manuscript. The authors offer a reassuring and informative response to my comments in respect of leader/follower roles and the question of teaching that had also been raised by Referee #3. The revision is aided by the way the authors have utilized the opportunities for a somewhat extended treatment of findings and additional citations, including those taking account the earlier referees' comments.

I have just a few minor comments urging very minor revisions as follows, by line number.

84-89 – Are predator identities known for any of these cases? If so I suggest adding in that information here.

94 – I suggest it would be appropriate to replace 'young' with 'surviving young'?

118 – I suggest replace 'the total amount of 32 registered steps' with 'the total number of 32 registered steps', or simply 'the 32 registered steps'.

191 – I suggest replace 'Being that' with 'Given that'.

END

Reviewer #4:

Remarks to the Author:

The manuscript, 'Paternal transmission of migration knowledge in a long-distance bird 1 d migrant' by Patrik Byholm et al. reports the results of a most interesting and instructive study of autumn migration in Caspian terns. The authors have shown that first-years migrants stayed close to one of the parents (biological parents or foster parents) at all times, and this bond did not dissipate until they reached their wintering grounds. All young that lost contact with their parent died. Solo-migrating adults migrated faster than did adults accompanying young, suggesting that adults carry a cost when they accompany first-time migrants. During their first solo migration, subadult Caspian terns remained faithful to routes they took with their parents as young, suggesting that information gained during their first migration with an adult is paramount for their subsequent migratory journeys.

During the previous rounds of reviewing, one of the referees was very cautious about the authors using the word 'teaching' when referring to the behaviour of adult Caspian terns that accompany the(ir) young and thus migrate more slowly than the parent that migrated solo. I would say that the

discussion whether or not this behaviour by adult Caspian terns might be referred to as 'teaching' is exciting but looks very scholastic to me.

I think that this is a very sound study, with proper methods used and with the conclusions fully justified by the results obtained.

Reviewer #5:

Remarks to the Author:

This is a highly interesting study that demonstrates the mechanism of parental transmission of migratory knowledge in a bird species (Caspian Tern) that habitually migrates in small parties of (typically, but not always, related) adults and young. Although the sample size is comparatively small (due to challenges of collecting long-term tracking data on multiple individuals migrating together), the results are convincing, and the authors are to be congratulated on a fine study. The paper has already been through one round of revision, and although I did not see the earlier draft, it appears that the authors have done a good job of dealing with the comments from the first round of review and improving the paper via the revision process. I find the study convincing, and generally well written, so my comments are all relatively minor.

1. In the abstract, the statement is made that "All young that lost contact with their parent died." Without reading the full paper, one would get the impression that "all" refers to many individuals, and there is no indication of the timespan of this effect - in reality only 4 birds died in this manner, and all either immediately before migrating or in the early stages. I think the statement should be modified to better reflect the results.

2. Line 64: "adult" should read "adults".

3. Lines 104-105: It is stated that the adult and young birds migrated "close together", with a mean difference in location of 0.6 km (plus/minus 2 km). This actually seems quite far apart if it is a true reflection of the inter-individual distance - can they see each other remain in contact at distances of e.g. 500 - 1500 m apart? This seems too far to me. Is it likely the real difference is smaller, but tag-position accuracy only allows this level of precision? If so, this should be stated. If not, please comment on the ability of birds to remain in contact while separated by distances of 1 km or more.

4. Line 109: Following from previous point, here it is stated that parents and young spent more time apart after arrival at winter site than during migration, but the actual distance between individuals was a mean of 1.1 (plus/minus 3.7 km) which overlaps with the inter-individual distance when migrating.

5. Figure 1. I found the symbols used rather confusing, and the legend was not as clear as it might have been. I think some more work needs to go into explaining the tracks and individuals in Fig 1. Also, the dates referred to in the legend do not appear on the figures.

6. Lines 179-181: Here it is stated that "In several cases, when naïve young broke away from the parent at advanced stages of migration, the young continued to migrate to other (more distant) wintering sites (Fig. 1c, Fig. 2, Supplementary Fig. 1) using highly goal-oriented flight behaviour." Does this not contradict the previous statements that all young which separated from parents during the migration died from predation? This needs clarifying. Also, I cannot see the young separating from adult in Fig 1c, as claimed.

7. Line 191: "Being" better as "Given".

Jason Chapman

REVIEWERS' COMMENTS

Reviewer #2 (Remarks to the Author):

In my (Referee #2) comments on the submission pre-dating the present one I had said that “This article offers what I judge are two interlinked ‘firsts’ that argue for a positive recommendation for publication in this journal” and I outlined those two, but then added a moderately substantive list of my reservations, largely concerning interpretations and citations to appropriate literature. This can all be seen in the Response to Referees document accompanying the present submission.

The rationale for my earlier positive recommendation stands. I judge that the authors have done a thorough and helpful job of responding to all of my earlier requests for revisions and I am satisfied with the resultant, present manuscript. The authors offer a reassuring and informative response to my comments in respect of leader/follower roles and the question of teaching that had also been raised by Referee #3. The revision is aided by the way the authors have utilized the opportunities for a somewhat extended treatment of findings and additional citations, including those taking account the earlier referees’ comments.

I have just a few minor comments urging very minor revisions as follows, by line number.

84-89 – Are predator identities known for any of these cases? If so I suggest adding in that information here.

Yes, it is known and we have now added this information to the text (L86-89).

94 – I suggest it would be appropriate to replace ‘young’ with ‘surviving young’?

Done

118 – I suggest replace ‘the total amount of 32 registered steps’ with ‘the total number of 32 registered steps’, or simply ‘the 32 registered steps’.

Done.

191 – I suggest replace ‘Being that’ with ‘Given that’.

Done.

END

Reviewer #4 (Remarks to the Author):

The manuscript, *Paternal transmission of migration knowledge in a long-distance bird 1 d migrant* by Patrik Byholm et al. reports the results of a most interesting and instructive study of autumn migration in Caspian terns. The authors have shown that first-years migrants stayed close to one of the parents (biological parents or foster parents) at all times, and this bond did not dissipate until they reached their wintering grounds. All young that lost contact with their parent died. Solo-migrating adults migrated faster than did adults accompanying young, suggesting that adults carry a cost when they accompany first-time migrants. During their first solo migration, subadult Caspian terns remained faithful to routes they took with their parents as young, suggesting that information gained during their first migration with an adult is paramount for their subsequent migratory journeys.

During the previous rounds of reviewing, one of the referees was very cautious about the authors using the word 'teaching' when referring to the behaviour of adult Caspian terns that accompany the(ir) young and thus migrate more slowly than the parent that migrated solo. I would say that the discussion whether or not this behaviour by adult Caspian terns might be referred to as 'teaching' is exciting but looks very scholastic to me.

I think that this is a very sound study, with proper methods used and with the conclusions fully justified by the results obtained.

We thank the reviewer for the supporting and encouraging words.

Reviewer #5 (Remarks to the Author):

This is a highly interesting study that demonstrates the mechanism of parental transmission of migratory knowledge in a bird species (Caspian Tern) that habitually migrates in small parties of (typically, but not always, related) adults and young. Although the sample size is comparatively small (due to challenges of collecting long-term tracking data on multiple individuals migrating together), the results are convincing, and the authors are to be congratulated on a fine study. The paper has already been through one round of revision, and although I did not see the earlier draft, it appears that the authors have done a good job of dealing with the comments from the first round of review and improving the paper via the revision process. I find the study convincing, and generally well written, so my comments are all relatively minor.

1. In the abstract, the statement is made that "All young that lost contact with their parent died." Without reading the full paper, one would get the impression that "all" refers to many individuals, and there is no indication of the timespan of this effect - in reality only 4 birds died in this manner, and all either immediately before migrating or in the early stages. I think the statement should be modified to better reflect the results.

Due to the length restrictions of the abstract (150 words used of 150 words allowed in the original version) there are limited possibilities for how much we can change the text. But we removed word "All" from the beginning of the sentence to try to avoid the impression the statement refers to many individuals (now 149 words used of 150 words allowed).

2. Line 64: "adult" should read "adults".

Corrected.

3. Lines 104-105: It is stated that the adult and young birds migrated "close together", with a mean difference in location of 0.6 km (plus/minus 2 km). This actually seems quite far apart if it is a true reflection of the inter-individual distance - can they see each other remain in contact at distances of e.g. 500 - 1500 m apart? This seems too far to me. Is it likely the real difference is smaller, but tag-position accuracy only allows this level of precision? If so, this should be stated. If not, please comment on the ability of birds to remain in contact while separated by distances of 1 km or more.

Thank you for pointing this out. Rather than changing our text here, we have added some more text to the methods-section clarifying this (L248-253). As we describe in the methodology-section, the pairwise parent-young distances (ΔD , $n = 21,264$) are calculated between fixes ≤ 20 min apart in time. Moreover, the internal clocks of the GPS-tags are not likely to be in perfect synchrony (and also the accuracy of the GPS-positions are likely to have an error magnitude). Together with our own visual field observations that adult and young Caspian terns migrating together typically are close to each other, this suggests the ΔD -values are likely to be somewhat inflated (although it is impossible to know exactly how much/little). Nevertheless, since the measurement error does not change temporally, ΔD -values are informative of how the distances between

members of adult-young pairs changes with advancing season (Fig. 2) even if the exact distance at a specific time is impossible to measure precisely.

4. Line 109: Following from previous point, here it is stated that parents and young spent more time apart after arrival at winter site than during migration, but the actual distance between individuals was a mean of 1.1 (plus/minus 3.7 km) which overlaps with the inter-individual distance when migrating.

We have now specified that the values refer to mean \pm s.d. (not e.g. mean \pm s.e. or CI) on L106, i.e. overlap as such is not indicative of non-significance. Our main point is not to compare differences in pairwise distances as compared between categories (during migration/after arriving to the parent's wintering location/after splitting up), but rather to present a summary of the data. We present the results from break-up analyses in Fig. 2.

5. Figure 1. I found the symbols used rather confusing, and the legend was not as clear as it might have been. I think some more work needs to be done into explaining the tracks and individuals in Fig 1. Also, the dates referred to in the legend do not appear on the figures.

We thank the reviewer for bringing our attention to this. The reference to dates was an error that resulted from us failing to correctly update the legend during previous re-drafting of the text. We have now corrected the legend of Fig. 1 what comes to this matter. We also specified the explanation that large intact/split circles indicate geographical locations of tracked birds, as identified by their respective colours, when the latter/single young started to migrate. I.e. by inspecting where these symbols are located in relation to each other one gets a "temporal snapshot" of the where the other family members are when the last family members started to migrate (see also Supplementary Figure 1).

6. Lines 179-181: Here it is stated that "In several cases, when naïve young broke away from the parent at advanced stages of migration, the young continued to migrate to other (more distant) wintering sites (Fig. 1c, Fig. 2, Supplementary Fig. 1) using highly goal-oriented flight behaviour." Does this not contradict the previous statements that all young which separated from parents during the migration died from predation? This needs clarifying. Also, I cannot see the young separating from adult in Fig 1c, as claimed.

Thank you for highlighting this. What we refer to here is what takes place after the breakup of the parental-young bond, i.e. after the young have become independent, i.e. it does not contradict the previous statement. To clarify this further we have now added the words "after becoming independent" to L181. In Fig. 1c it is Young 2 (white diamonds) we refer to in the text: it separates from its parent (which stays wintering in the Nile delta; marine blue track + diamond) to continue to Lake tana in Ethiopia.

7. Line 191: "Being" better as "Given".

Corrected.

Jason Chapman